# Improvement of the Treatment Performance of the Conventional Wastewater Treatment Plant: A Case Study of the Central Wastewater Treatment Plant in Ulaanbaatar, Mongolia

Boldbaatar Nyamjav [1], Ayurzana Badarch [1] , Chuluunkhuyag Sangi [2] and Ganbaatar Khurelbaatar [3,*]

1. Department of Engineering Facilities, School of Civil Engineering and Architecture, Mongolian University of Science and Technology, Ulaanbaatar 14191, Mongolia
2. Department of Environmental Engineering, School of Civil Engineering and Architecture, Mongolian University of Science and Technology, Ulaanbaatar 14191, Mongolia
3. Centre for Environmental Biotechnology, Helmholtz Centre for Environmental Research—UFZ, 04318 Leipzig, Germany
* Correspondence: ganbaatar.khurelbaatar@ufz.de; Tel.: +40-341-235-1842

**Abstract:** This study is carried out within the framework of the requirements of ensuring reliable operation of the Central Wastewater Treatment Plant of Ulaanbaatar to increase the treatment performance and mitigate the adverse impacts of the current poor performance on nature and ecology. The dynamic of the current influent was studied in order to find the best-suited solution as a means of improvement for the WWTP. As an immediate and fast-to-build solution, an equalization tank and pre-aeration facilities were proposed. The positive effect of the facilities on the biological treatment stage and the overall treatment performance was estimated both theoretically and empirically. The results showed that the overall performance of the treatment for $BOD_5$ and TSS will be increased by up to 30% and 10%, respectively. Furthermore, the high fluctuation problems the central WWTP in terms of hydraulic and pollution load will be positively affected by the intervention, eliminating the main cause of the current poor treatment performance.

**Keywords:** wastewater; total suspended solids; biochemical oxygen demand; equalization tank; pre-aeration



## 1. Introduction

The central wastewater treatment plant in Ulaanbaatar is facing major challenges, both in terms of treatment performance and structural aging. Similar to many other wastewater treatment plants built in Mongolia in the 1970s and 1980s, the Ulaanbaatar Central Wastewater Treatment Plant is nearing the end of its 40-year life cycle. The gradual increase in population in Ulaanbaatar and the sudden increase in the industry mean that the treatment plant is overloaded at certain times. These overloads have recently been occurring more frequently than ever, resulting in the entire system not being able to perform normally.

The centralized WWTP of the city of Ulaanbaatar was built between 1969 and 1986 stage by stage with the initial intent of treating only domestic wastewater and thus included basic mechanical and biological treatment stages [1]. Since its completion, the WWTP has not receive major restoration except for minor repairs and maintenance work. This caused a significant alteration of the equipment and facilities, which in part resulting in deterioration of the treatment performance [2]. Another challenge facing the wastewater infrastructure in Ulaanbaatar is facing is the sudden increase in both hydraulic and pollution loads due to the demographical change and the new establishment of small and medium industries within the city boundary [3]. In the current situation, where the treatment plant is constantly overloaded both hydraulically and in terms of pollution rate,

overall performance decreases, so the wastewater quality is 4 to 8 times higher than that allowed by local regulations [2].

These challenges make an intervention in terms of improvement of the treatment performance a necessity. These interventions can include the construction of a completely new wastewater treatment plant or solution, which is more immediate and on a contemporary basis. In terms of the construction of a new WWTP, it might be time-consuming and costly due to the high investment and O&M costs associated with the cold climate of the region. The objective of this study is (a) to evaluate the current treatment performance of the central wastewater treatment plant in Ulaanbaatar, (b) to identify the main causes of performance deterioration, and (c) to propose and investigate solutions that can be immediately implemented to improve the current situation. As an immediate means of alleviating the current condition, two additional infrastructures were proposed and their effects on the treatment performance were studied.

## 2. Materials and Methods

### 2.1. The Central Wastewater Treatment Plant

The Central Wastewater Treatment Plant (CWWTP), located southwest of Ulaanbaatar city in the Songinokhairkhan district, was commissioned in 1969 and built in several stages until its completion in 1986. As the country's largest wastewater treatment plant, CWWTP occupies an area of 176,271 $m^2$ and currently has the capacity to treat 170,000 $m^3$ of wastewater per day (Figure 1).

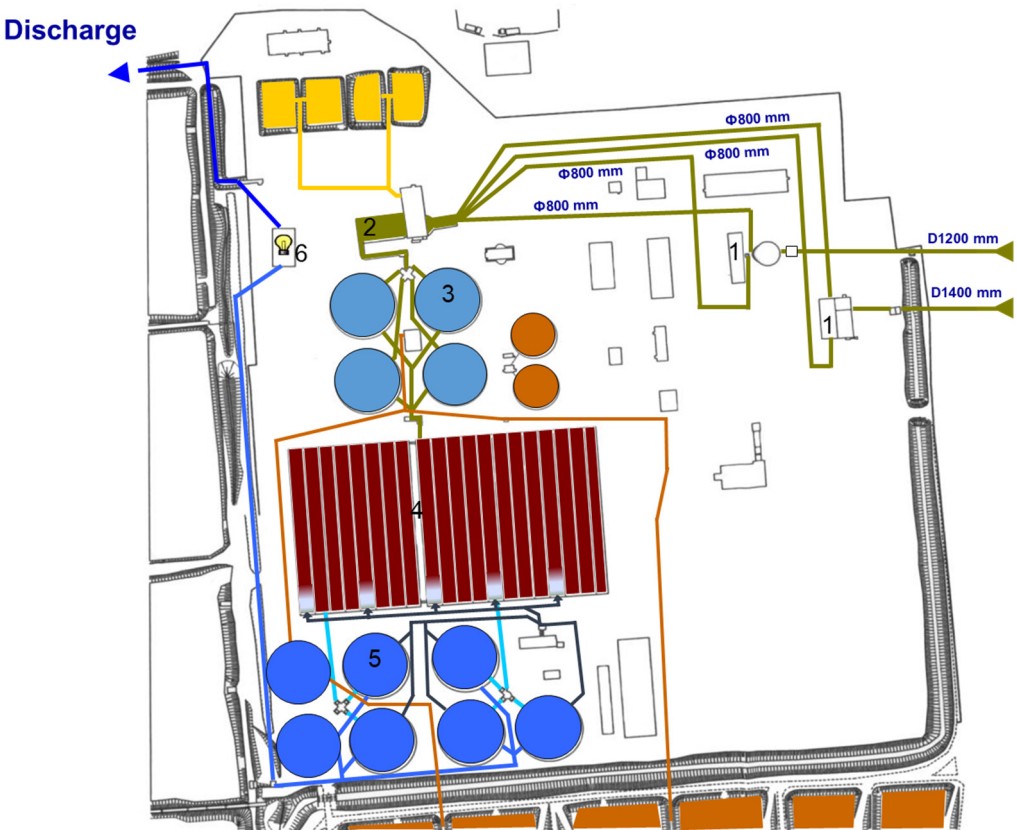

**Figure 1.** Diagram of the CWWTP in Ulaanbaatar. 1. Screening, 2. Sand trap, 3. Primary clarifier, 4. Aeration tank, 5. Secondary clarifier, 6. Chlorination.

Initially, the CWWTP was designed to treat only domestic wastewater with maximum concentrations of $BOD_5$: 250 mg/L, TSS: 250 mg/L, and COD: 600 mg/L. However, today it receives both domestic and industrial wastewater [3]. The initial design capacity of 230,000 $m^3$ [4] was gradually reduced due to the deterioration of the equipment and

infrastructure to 170,000 m$^3$/day of the current capacity. It was observed that all CWWTP facilities were filled and started to overflow when the hydraulic load reached 176,000 m$^3$/day. In such situations, wastewater accumulates in the main trunk lines consisting of 1200 mm and 1400 mm diameter pipelines and creates up to 5 km congestion from the inlet of the CWWTP towards the city. The amount of raw wastewater that accumulates in the trunk line reaches 10,000 to 12,000 m$^3$ and thus creates a risk of direct discharge of raw wastewater into the environment [5]. In recent years, during storm events, the hydraulic rate even reached 230,000 m$^3$, and it is expected that the hydraulic rate will increase in the future [5].

In addition to hydraulic load, there has been a sudden increase in pollution concentration in recent decades (Figure 2).

Furthermore, the CWWTP has been subject to seasonal changes, which could have a negative impact on treatment performance. For example, many residents of Ulaanbaatar city usually go to the countryside during the summer season (months 6 through 9); therefore, the amount of domestic wastewater is reduced during this period. On the contrary, during the winter period, the population and industry activity increase within the city, increasing the hydraulic and pollution load. The water quality data of the last decade shows that the treatment efficiency for BOD$_5$ and TSS was approximately 65–75% and 80–89%, respectively [2,5]. The mean annual concentrations for BOD$_5$ and TSS were 118–149 mg/L and 115–173 mg/L, respectively. This means that the concentration of BOD$_5$ is 5 to 8 times higher than the allowed concentration, while the TSS concentration is 4 to 6 times higher (Figure 3).

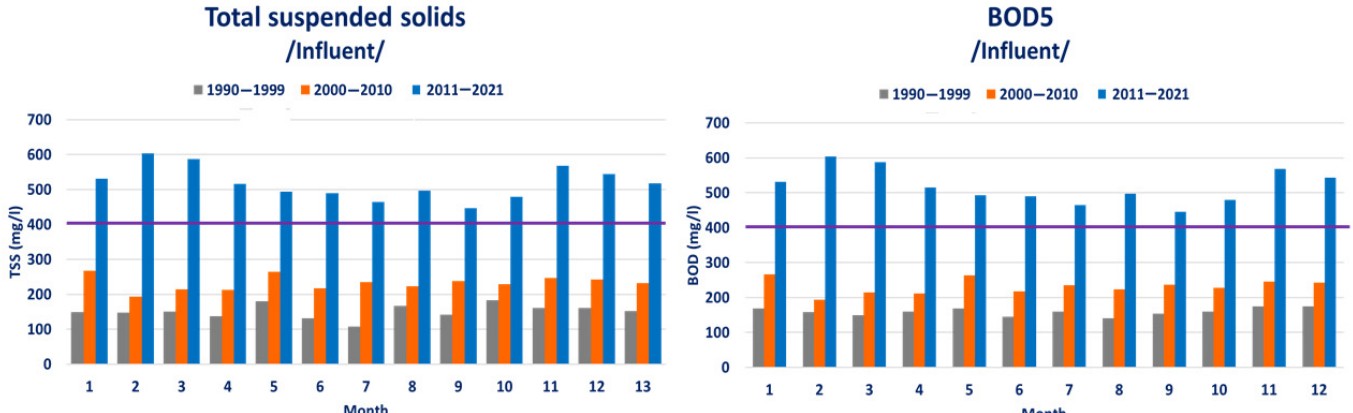

**Figure 2.** Mean monthly influent concentrations (the concentration of 400 mg/L indicated in the standard MNS 6561:2015 [6] is expressed by a straight line).

The main factor that causes this low treatment efficiency is found to be the high fluctuation of pollutants in the influent. The efficiency of treatment was observed to increase with the decreased concentration of pollution, clearly suggesting that the current state of the CWWTP is overloaded both in terms of pollution and hydraulic [2,5]. Another parameter that stood out in the observation was the sludge index in the aeration tank [2,5]. The sludge index determines the settling ability of activated sludge, and the settling process favors a sludge index of 60–150 mL/g [7–9]. In certain more advanced WWTPs, the sludge index is kept even lower at 50–100 mL/g [10,11]. As for the CWWTP, this index was recorded as 330–1040 mL/g between 2019 and 2022.

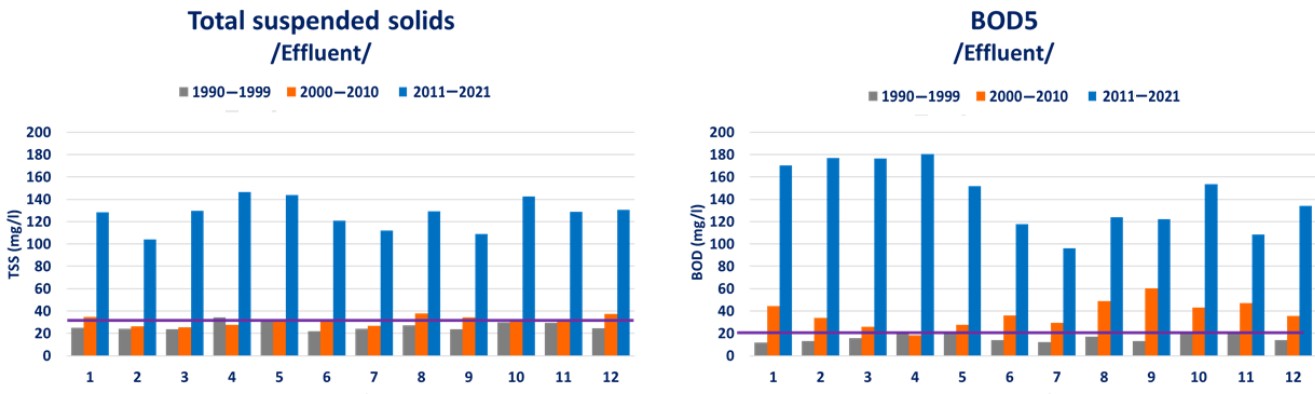

**Figure 3.** The mean monthly effluent concentration (the allowed discharge concentration of TSS: 30 mg/L, BOD$_5$: 20 mg/L in accordance with the national the standard MNS 4943:2015 [12] is expressed by a straight line).

The high level of pollution and the high sludge index are assumed to be attributed to the high amount of industrial wastewater [5]. Due to the highly polluted influent in recent years, especially in the case of industrial wastewater that does not comply with the national standard [5], coupled with aging equipment and facilities straining under extreme climatic conditions, the CWWTP urgently requires an immediate solution to ensure operation, which can guarantee the effluent quality according to the national standard [5] all year round.

### 2.2. Detailed Assessment of Current Inflow Dynamic

To study the dynamic of current inflow in terms of hydraulic and pollution load at the CWWWTP, several field and lab experiments were carried out. Hourly inflow measurements were carried out on 15 January, 15 April, 15 July, and 15 October 2021 for 24 h in each measurement event. An Siemens MAG 5000 electromagnetic flow meter (Siemens AG, Nürnberg, Germany) was used at the main entry point of the CWWTP to measure the hourly and daily amount of wastewater inflow.

Additional sampling and monitoring campaigns were conducted on 2, 6, and 26 October 2021. After 72 h, samples were collected from the influent and analyzed in the water quality monitoring lab at the CWWTP [5,13]. The parameters analyzed were BOD$_5$, TSS, pH, and DO in accordance with the Mongolian national standard MNS; sampling procedure: MNS ISO5667-10:2001 [14], BOD$_5$ analysis: MNS ISO 5815-1:2015 [15], MNS ISO 5815-2:2015 [16], and TSS analysis: MNS 11923:2001 [17]. Together with the influent and effluent quality data provided by the CWWTP, the results were then used for the design and calculation of the necessary interventions and their potential effect on the current treatment performance.

### 2.3. Proposed Solution

Compared to many other countries that started WWTP technologies a century ago and therefore have more experience, Mongolia is facing the challenge of sustainably renovating the WWTP for the first time. The same applies to the state of the research regarding WWTP technologies in Mongolia. As an improvement and immediate alleviation of the current condition of the CWWTP in Ulaanbaatar, many technical solutions can be proposed, such as building a new wastewater treatment plant and implementing pre-treatment and post-treatment stages, since the current treatment plant is already reaching the end of its life cycle. Building a new WWTP is always associated with high costs [5], and the completion may take years, while more contemporary and immediate solutions may be more suitable.

Some researchers have suggested implementing a tertiary treatment stage as a means to improve the current situation [18]. However, the tertiary treatment would not address the issue of the high fluctuation of the influent in terms of quality and quantity. Instead,

an intervention is necessary to improve the performance of the primary treatment by controlling the hydraulic and pollution loads and enhancing the $BOD_5$ and TSS. Without stabilizing primary treatment, further treatment such as tertiary treatment are considered to be of minimal effect and expensive [19].

In general, to increase the performance of the primary treatment, preliminary aeration and/or a method of aeration by adding activated sludge to the influent are used prior to sending wastewater to the primary settling tank, and these methods are called pre-aeration and/or bio coagulation [20–23]. In both cases, the sedimentability of solids is improved by changing their hydraulic grain size [20,22]. With the use of these facilities, it is possible to increase the removal efficiency of TSS [20–23] in the primary, secondary, and tertiary stages. The main differences between pre-aeration and bio coagulation are that pre-aeration speeds the sedimentation process up while during bio coagulation, biochemical oxidation of some easily oxidized solutes occurs in addition to the physiochemical and biochemical processes [20,22]. Therefore, the following two interventions are proposed in this study and the potential effects are studied. Although the tertiary treatment proposed by some researchers and the construction of new treatment facilities and chemical treatment have disadvantages such as high cost and increases in the operating costs, the method proposed and tested in this study is the simplest and easiest way to improve the effectiveness of current WWTP treatment [5,22,23].

### 2.3.1. Equalization Tank

In order to cope with the high fluctuations of the inflow in terms of both hydraulic and pollutant load, an equalization tank was proposed. The equalization tank will allow a more uniform influent, both hydraulic and concentration-wise, eliminating the fact that the CWWTP must receive maximum flow in a short period of time [24–26]. The equalization tank also provides additional capacity in case of storm events, so that the risk of discharging raw wastewater as a result of combined sewer overflow will be reduced. The equalization tank is usually designed for pollution concentrations/TSS and $BOD_5$/greater than 500 mg/L, and parameters such as pollution fluctuations and the hourly changes in the concentration of pollutants in the wastewater received daily are required for relevant design [7–9].

### 2.3.2. Pre-Aeration

The equalization can be combined with a pre-aeration tank. When the concentrations of $BOD_5$ and TSS both exceed 100–150 mg/L, it creates difficulties in future wastewater treatment processes [7,8,27]. Pre-aeration is generally used to support the primary settling tank and the biological treatment process in the treatment plant, and it can also be used to prevent odor problems. The pre-aeration technology is generally recommended for use in the following cases:

- To increase the performance of the primary settling tank if the concentration of pollutants in the primary settling tank is greater than the total suspended solids 300 mg/L [7,8,27].
- In the case of mixing industrial wastewater that adversely impacts the biological treatment process [7,8,27].

Pre-aeration is usually carried out in a special pre-aeration tank or in the supply channel before the primary settling tank or in a special facility built for this purpose [22,23]. The speed of this process depends on the amount of free oxygen in the wastewater itself and the aeration process [22,23,28]. At this stage, all types of organic substances are oxidized with the help of oxygen and mineralizing microorganisms [22,23,28]. In the course of aeration, agglomeration and/or coagulation of the smallest particles of undissolved pollutants in the wastewater occurs, the density of which is slightly different from the density of water itself. As a result, these particles change their hydraulic grain size and become able to settle more quickly when entering the next settling facility [7,8].

The aeration is usually up to 30 min, and the amount of air given to the wastewater is generally 0.5 m$^3$ of air per 1 m$^3$ of wastewater; however, the amount of aeration can be increased to improve the effectiveness of treatment [22,23]. The purpose of pre-aeration at the CWWTP is to reduce the sludge index by exposing the raw wastewater to aeration for 30 min before it enters the primary clarifier. However, in this study, we did not propose creating sludge through bio coagulation due to the fact that additional sludge will have a negative impact on the subsequent treatment chain due to the current high sludge index [5]. Pre-aeration can be performed structurally in a primary settling tank or in a standalone facility.

*2.4. Design of the Equalization Tank and Pre-Aeration and the Estimated Effect on the Treatment Performance*

The equalization tank should be designed along the settling tank or alongside the settling tank to facilitate its use. The pace at which wastewater arrives at the treatment process can fluctuate substantially during the day, so it is necessary to equalize the flow before feeding it to the various treatment phases. The process of controlling flow velocity and composition is known as flow equalization and is used to dampen extreme variations in the flow and water quality in many municipal and industrial treatment procedures. The controlled flow of the hydraulic load minimizes the adverse effects on the performance of the treatment in the biological stage and therefore increases the treatment performance [29–31]. The larger the relative volume of the equalization tank or basin, the more stable the concentration of pollutants will be throughout the day. When evaluating the performance of the treatment plant, it is often useful to be able to predict the impact of flow equalization on the concentration of pollutants [32].

In this study, various sizes of equalization tanks were assessed for their effect on the average concentration of BOD$_5$ and TSS in the outlet. The hourly average concentration of the pollutants was estimated using Equation (1) in accordance with [32]. First, we calculated the average hourly flow rates and used those to determine the average volume of flow entering the equalization basin each hour. The overall volume of water in the tank at any given time was then computed by subtracting the average daily flow rate from the fluctuating hourly flow rate.

$$BOD_{tank.t} = \frac{BOD_{in.t}V_{in.t} + BOD_{in.t-1}V_{in.t-1}}{V_{tank.t-1} + V_{in.t}} \tag{1}$$

where $BOD_{in.t}$: BOD concentration of influent:

- $BOD_{tank.t}$: BOD concentration of the effluent of the equalization tank.
- $V_{in.t}$: Hourly volume of the influent.
- $V_{tank.t}$: is the volume of wastewater stored in the tank at time $t$ or $t-1$.

Based on the dynamics of the hydraulic and pollution load of the influent assessed in this study, various sizes of the equalization tank have been taken into account, and the effluent concentrations were estimated in Equation (1) [32].

Calculations of the pre-aerator are given for the case of wastewater treatment in aeration tanks. The pre-aeration volume is determined by the calculated flow rate based on the duration of aeration T = 10–30 min [23,28]:

$$W_{tank.aera} = \frac{Q_{hour.max} \times T_{aera}}{60} = \frac{9278 \times 30}{60} = 4639 \approx 5000 \text{ m}^3 \tag{2}$$

The pre-aeration tank was designed to be 5000 m$^3$ (Equation (2)). In order to study the effect of the pre-aeration tank, the sludge index and concentrations of the pollutants TSS and BOD$_5$ were calculated according to Seidel [28]. To validate the estimated results, a lab experiment was carried out. The samples were taken from the outlet of the sand trap (Figure 1) on 15 June 2021 and 24 August 2021 at 9:00, 11:00, 13:00 and 15:00. The sampling

was performed according to the methods and techniques specified in the domestic standard MNS ISO 5667-10:2001 [14].

The samples were then tested in the laboratory experiment. Twenty liters of the wastewater sample were poured into a 50 L glass container and continuously aerated for 30 min. The aeration rate was 1: 1, which means that 20 L of air was injected into the 20 L wastewater sample within 30 min. The particle size of the air bubble was 0.5 to 1.0 mm. The wastewater was analyzed before and after the experiment for $BOD_5$ and TSS according to the local standard [15–17] (Figure 4).

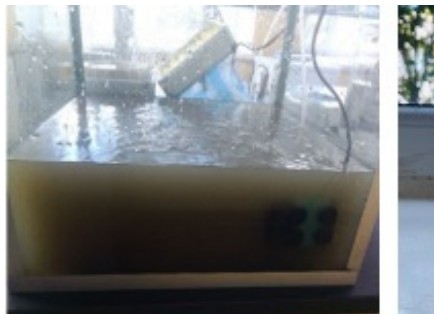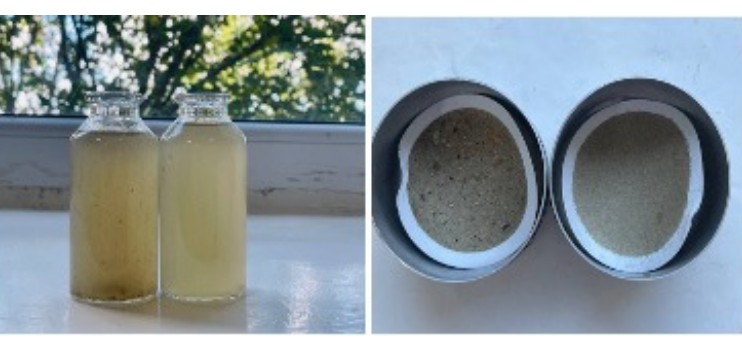

**Figure 4.** Lab experiment: Pre-treatment, samples before and after pre-aeration. TSS content before and after the experiment.

## 3. Results and Discussion

### 3.1. Inflow Dynamic of the CWWTP

A detailed assessment of inflow showed the dynamic of the influent in terms of hydraulic and pollution load. The amount of wastewater varied between 4.625 and 8.395 m$^3$/h (Figure 5) in general. Furthermore, 160.716 m$^3$/day was received on 15 January 2021, while 150.589 m$^3$/day, 176.679 m$^3$/day, and 191.236 m$^3$/day of wastewater were received on 15 April, 15 July and 15 October, respectively.

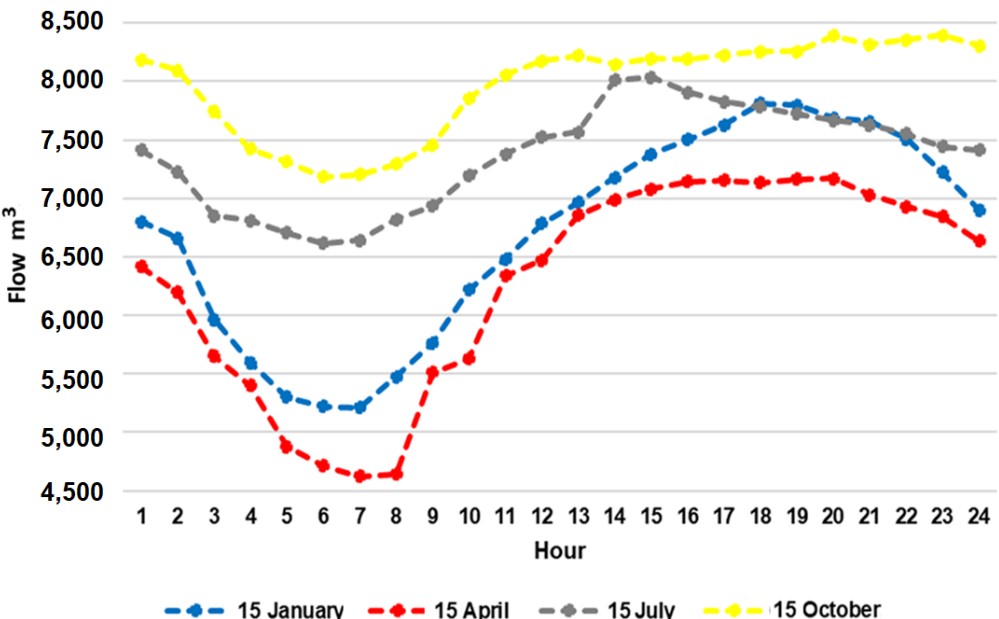

**Figure 5.** Influent volume of wastewater.

Additionally, it was reported by the local data of CWWTP that the lowest amount of wastewater of 2.890 m$^3$/h was received on 7 April 2021 between 06:00 and 07:00 o'clock,

while the highest amount of wastewater of 9.278 m³/h was received on 4 October 2021 between 22:00 and 23:00 o'clock.

The amounts of sewage entering the dual inlet lines of the CWWTP from the line coming from the industrial area and the municipal or domestic sewage line are almost equal. For example, the amount of sewage from the industrial zone accounted for 53%:47% in 2020 and 51%:49% in 2021, respectively.

The statistical data used and supplied by the CWWTP are expressed as annual, monthly and daily averages. In order to precisely estimate the effectiveness of the equalization tank and the pre-aeration, it was necessary to measure the amount of wastewater and pollution concentrations on an hourly basis. The CWWTP has two inlet lines: (1) the 1200 mm diameter line/city or domestic sewage line and (2) the 1400 mm diameter line/wastewater line coming from the industrial area [1].

The concentration of TSS and $BOD_5$ varied between 177 and 2.365 mg/L and 100 and 1.214 mg/L, respectively (Figures 6 and 7) [5,13].

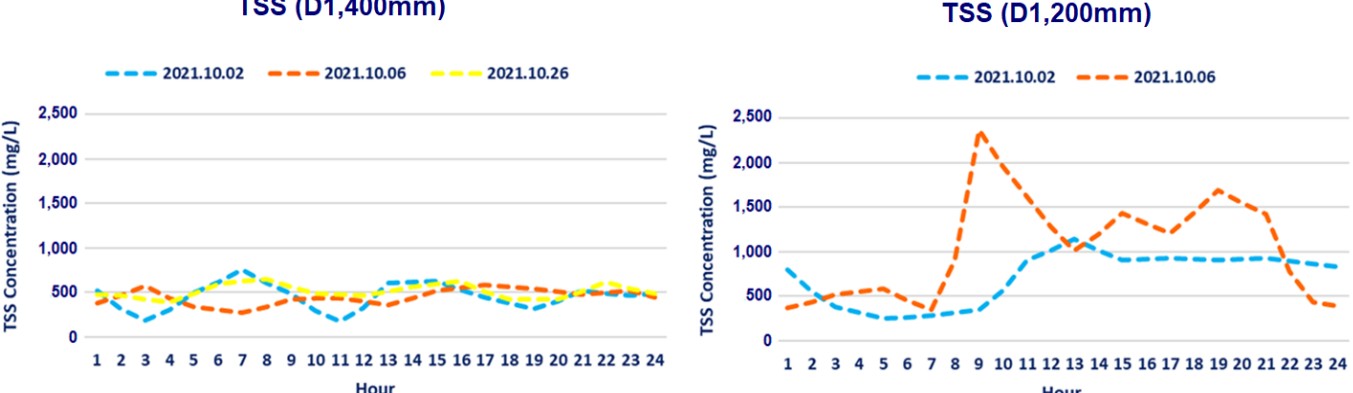

**Figure 6.** Hourly concentration of TSS at the inlets.

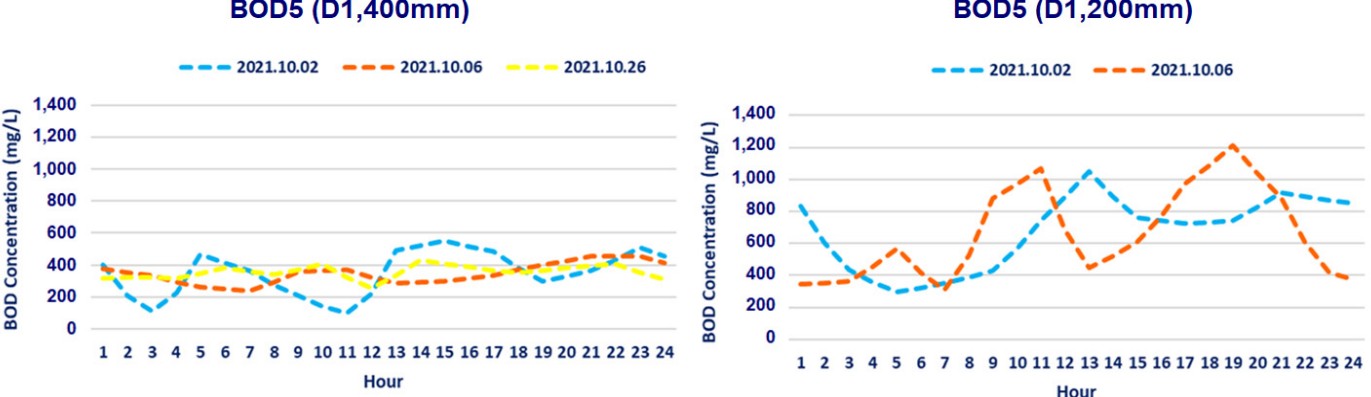

**Figure 7.** Hourly concentration of $BOD_5$ at the inlets.

The amount and pollution rate of wastewater received by the CWWTP vary from hour to hour, and this variation and fluctuation can be eliminated as a result of the design of an equalization tank.

### 3.2. Effect of Equalization Tank and Pre-Aeration

The tank for equalization was determined from an inflow mass hydrograph of the hourly fluctuations for a typical daily wastewater flow (Figure 8).

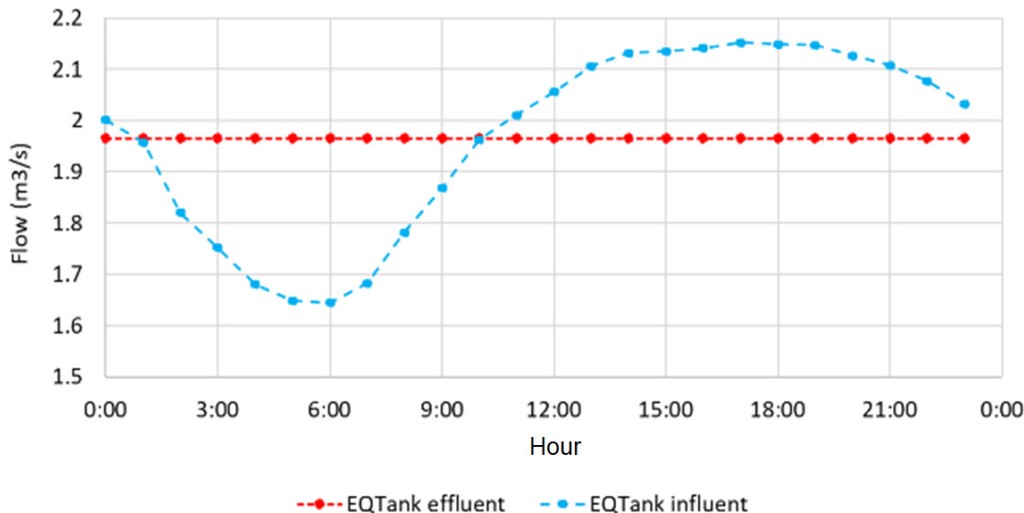

**Figure 8.** Equalization of flow rate.

The results showed that the tank capacity has a big impact on changes in the concentration of pollutants. The following figures (Figures 9 and 10) show the effect of equalization tanks with a capacity of 10,000–100,000 m$^3$ on the change in the concentration of wastewater pollution [5].

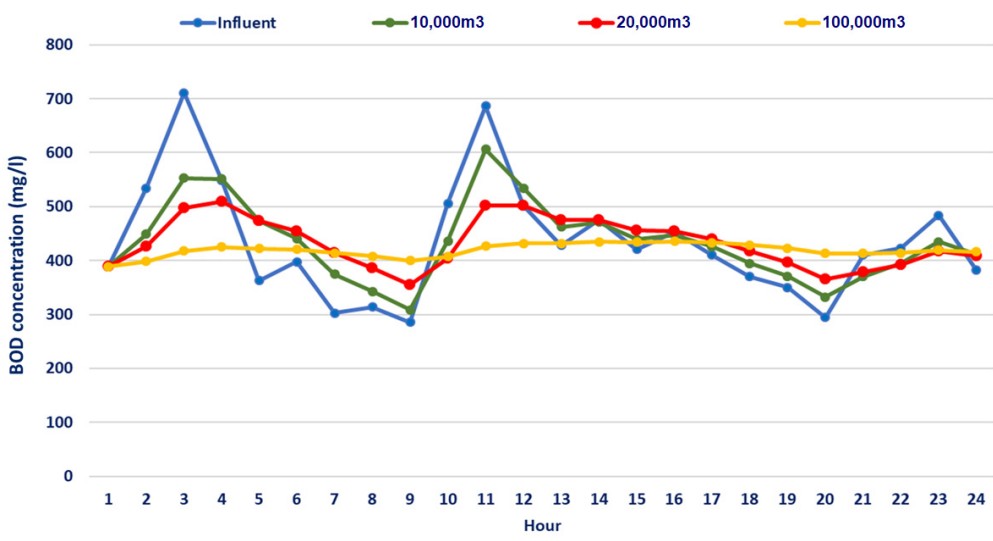

**Figure 9.** Effect of the equalization operation on the BOD concentration on an hourly basis.

### 3.3. Effect of Pre-Aeration

An experiment was conducted to identify the effect of pre-aeration on wastewater treatment and how the effectiveness of pollution treatment changes after 30 min from the initial concentrations. The experiments were carried out on 19–30 June and 24–28 August 2022 and the results showed that the concentration of suspended matter was reduced by 27–57% and that of BOD$_5$ was reduced by 17 to 33% (Figure 11).

Assuming that the current efficiency of treatment after the primary settling tank is 20–30% for BOD$_5$ and 40–50% for suspended particles [33], as a result of the introduction of the equalization tank and pre-aeration before the primary treatment, the effectiveness of treatment of the primary settling treatment improves by an additional 10–20% for both parameters.

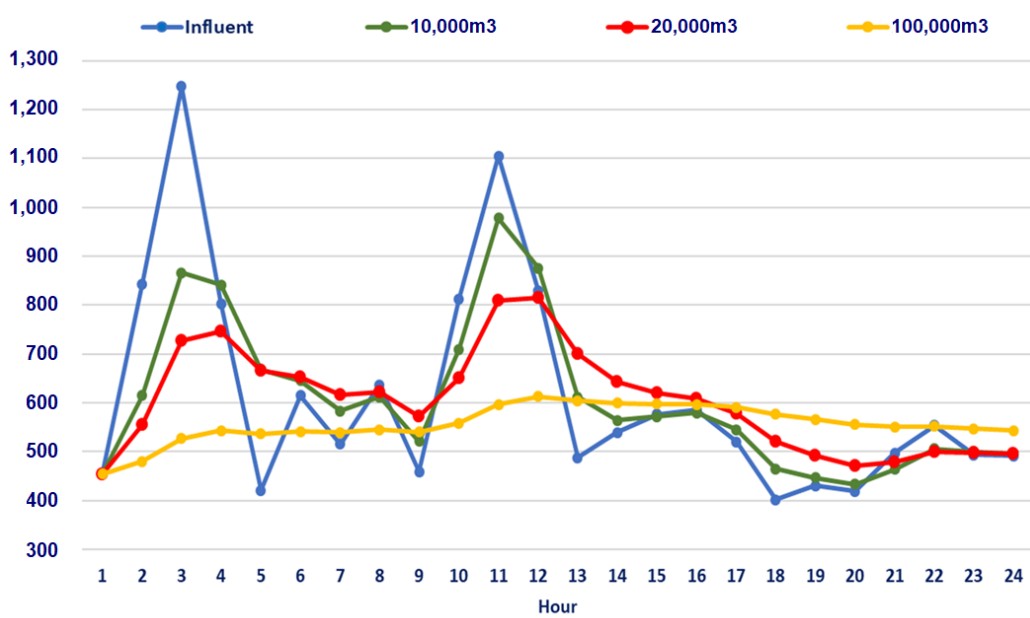

**Figure 10.** Effect of the equalization operation on the hourly TSS concentration.

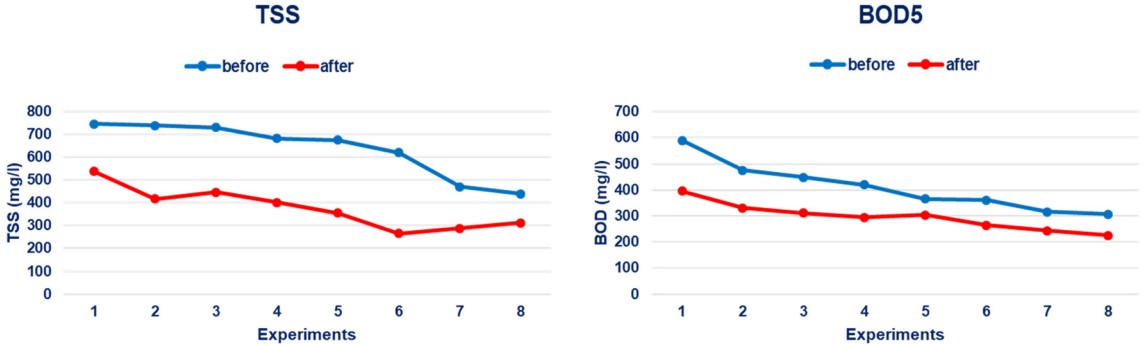

**Figure 11.** The effect on BOD5 and TSS.

The results of this study in terms of pre-aeration are similar to that reported by V.V. Kuibyshev [23,34]. In their study, the effectiveness of reducing the TSS increased by 30% compared to the ordinary settling unit, and for BOD, the effectiveness increased by 35%. In a primary settling tank with pre-aeration, the removal efficiency of TSS increased up to 65–70% and the concentration of $BOD_{20}$ decreased by approximately 15% [23,34].

Furthermore, the efficiency of equalization tanks has been reported to have a similar positive effect on treatment performance in general [29]. The high fluctuation of the influent in terms of both hydraulic and pollution load adversely impacts the further treatment chain. As a result of the introduction of an equalization tank, both the amount of water that comes out of the tank and the concentration of pollution will reach their average value, and their fluctuation will be eliminated, resulting in stabilization [30,31].

Long-term data analysis suggests that the treatment efficiency for $BOD_5$ was sufficient (88–93%) when the $BOD_5$ concentration was low in the influent (Figures 2 and 3). However, in the last 10 years, due to the fluctuations in hydraulic and pollution load, the above situation has been lost, and the effectiveness of primary treatment has decreased, resulting in the deterioration of the effectiveness of not only of biological treatment but also the overall treatment.

As a result of planning pre-aeration, it will be possible to meet the basic requirements of biological treatment (according to the domestic standard, the concentration of suspended matter and $BOD_5$ should be 150 mg/L [35]), and the effectiveness of primary treatment at the treatment plant can increase by 10% to 15% [23].

During the 30 min spent in the equalization tank, the wastewater is provided with continuous aeration, therefore, when entering the aero tank, the process of oxidizing organic pollution dissolved in the wastewater, which is the main process of biological treatment, will be more activated [5,18].

As a result of oxidation, part of organic pollution becomes water, carbon dioxide, nitrate, and sulfate ions, while the rest is formed into biomass. Furthermore, this biomass will be re-aerated in the aeration tank, and at that time, the amount of air in the wastewater composition will already be sufficient ($2 \, \text{mgO}_2/\text{l}$) [5,18].

Consequently, it has been theoretically estimated that it is possible to increase the treatment effectiveness of suspended particles by 5–10% and increase the treatment effectiveness of $BOD_5$ by 25–30% through the use of an equalization tank and pre-aeration [10,18].

## 4. Conclusions

In this study, an immediate intervention to improve the current and poor treatment performance of the Ulaanbaatar CWWTP has been studied. As the simplest and most viable option for alleviation, an equalization tank and pre-aeration technology have been proposed. Furthermore, considering the long-term data analysis on the treatment performance of the CWWTP, the proposed solutions are considered the best and fastest means of improvement compared to other measures such as building a new WWTP.

As a result of implementing the equalization tank and pre-aeration facility in the treatment technology, the treatment effectiveness after mechanical treatment increased from 34% to 63% for $BOD_5$ and from 68% to 86% for TSS, thus meeting the requirements of domestic standards. The effects of the facilities on mechanical treatment were estimated based on both theoretical and empirical studies, while the effect on the biological treatment was estimated [5].

The results of this study suggested that the performance of the primary treatment will increase, producing a positive impact not only on the performance of the next stage of biological treatment but also on the overall treatment performance. The current study is distinguished from other similar works performed in this field as it plans a pre-aeration stage based on the properties of the wastewater received by the CWWTP and the fluctuations and changes in hydraulic and pollution loads. The results showed that not only will the treatment performance of the CWWTP be improved but also its normal and reliable operation will be ensured [5].

**Author Contributions:** B.N. contributed to the conception of this study. C.S. and A.B. contributed to the conception of experiments and writing of the study. B.N. carried out the experiments and data analysis. B.N. and G.K. wrote the manuscript. All authors have read and agreed to the published version of the manuscript.

**Funding:** This research was funded by the Civil Engineering and Architecture school, Mongolian University of Science and technology (MUST).

**Institutional Review Board Statement:** Not applicable.

**Informed Consent Statement:** Not applicable.

**Data Availability Statement:** The raw and processed data supporting the conclusion of this study can be made available by the authors, without undue reservation. Further inquiries should be directed to the corresponding author, B.N.

**Acknowledgments:** The authors thank USUG for providing all data used in this study. The authors also acknowledge the Mongolian University of Science and technology for providing technical and financial support. Finally, we thank anonymous editors and reviewers for their valuable suggestions and comments that significantly improved the quality of the manuscript.

**Conflicts of Interest:** The authors declare no conflict of interest. The funders had no role in the design of the study; in the collection, analyses, or interpretation of data; in the writing of the manuscript; or in the decision to publish the results.

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
