# Peer review of "Improvement of the Treatment Performance of the Conventional Wastewater Treatment Plant: A Case Study of the Central Wastewater Treatment Plant in Ulaanbaatar, Mongolia"

_sustainability, doi:10.3390/su15054528_

Round 1

Reviewer 1 Report

Dear authors, 

I am quite reluctant to suggest the manuscript for publication due to some fundamental issues. If you feel you can answer back my issues properly (along with other reviewers' ), you are suggested to do so. Otherwise, you may withdraw and resubmit it later on. The topic is novel and interesting.

First and foremost, the title and the arguments presented at the beginning of the introduction are rather misleading. You are talking about problems under cold temperatures. I would like to point out that temperature does not affect directly the performance of the WWTP studied (or at least it is not the main problem you are facing) but it is actually the higher hydraulic and pollutant load, occurring more under winter seasons, that undermines the performance of the system. If you want to publish the paper, the first thing to do is to change drastically the title. I can also tell you that in a WWTP that I studied located in a tourist area, the higher pollutant load occurs in summer while in winter we don't face any issue about BOD or TSS. The leading cause of the bad system's performance is the overloading. Secondly, when talking about WWTP performance you only consider BOD and TSS while other important pollutants such as ammonium can be even more relevant, especially in winter as autotrophs do not work well. It's well know that winter-related WWTP problems are linked to increased effluent ammonium. Thirdly - and this the most sever issue I have - is the lack of evidences that the proposed interventions of equalization and pre-aeration tanks can improve the WWTP performance and the effluent quality. There is really no compelling evidence with this regard, I can only see better primary settling, but what about the overall WWTP performance? And what about ammonium? Don't you have regulations on effluent ammonium?

I must acknowledge the paper is well written in a very clear English., which is becoming to my experience a very rare case. So that's a very good point. Very few English flaws.

Detailed comments as follows: 

L123: are you sure that’s a good choice? What about the other days of the same season? Don’t you have at least an average flow rate value?

L97-101: Sludge VOLUMETRIC index: I don’t think it’s that bad. I don’t see that as a relevant reason for high TSS in the effluent.

L154-157: You need to elaborate upon those concepts a lot better than just simply including references. You have to explain what happens to the sedimentability of solids when you do pre aeration or when you add activated sludge. And you should be explaining this within L154 and 157, not just afterwards. Otherwise reader may lose focus. Furthermore, I may try to understand why pre aeration may be beneficial for primary settling but not for secondary or tertiary settling (L158-159).

Figure 11: it is not clear what the two blue and red lines are: are they influent and effluent from the preaeration tanks? Wasn’t the role of pre-aeration to increase solids sedimentability by biocoagulation? This is confusing

Figure 12: I don’t understand how these data were generated, being the influent concentrations of BOD and TSS the same before and after the intervention. Furthermore, when you talk about “effluent”, is that the effluent of the plant or the effluent from the primary settler? I guess it’s the effluent from the plant. If so, I honestly don’t see any advantage in applying pre-aeration or equalization.

L343: nitrite? What are you talking about? Usually very low concentrations of nitrite are expected in the effluent, unless severe malfunctioning occurs.

Reviewer 2 Report

Dear Authors

The paper it is interesting as it is increasingly important to find cost-effective ways to improve the efficiency and effectiveness of WWTPs in operation. The objectives are clearly defined in the last paragraph of the introduction section. However, the way in which the article is structured significantly interfered with its analysis and interpretation. In fact, part of point 2.1 of section 2 - Materials and Methods includes treatment and discussion of results so, in my opinion, it should be included in section 3 - Results and discussion. Also, when you tried to evaluate the effect of pre-aeration and the existence of an equalization tank on the efficiency of the WWTP, you did not clearly describe with sufficient detail the laboratory tests and, therefore, I did not understand which variables were tested in the 8 experiments you refer to in figure 8. It is important that you give enough detail so that other researchers are able to carry out the same research. I also think that in Materials and Methods section the statistical analysis of the data should be more detailed. The references do not appear in the text sequentially, for example reference 5 is only cited for the first time on page 5 of the article. In page 1, line 36 the name of the reference does not match reference 2 in the list of references. Also, the figures are not cited in the text sequentially. For example, they speak first of figure 3 and only later refer to figure 2. Page 3, third paragraph, when you refer to the sludge index units, shouldn't it be mL/g instead of mg/g? Dear authors, I think your paper has some potential, but you should carry out a major revision. It would also be important that you try to improve the statistical treatment of the data (for example, I think it would be important at least to present the standard deviation when you refer the mean values). I suggest you spend time with reorganize and improve the materials and methods and results and discussion sections in order to ensure a better understanding of the results obtained.

My best regards

Conceição Mesquita

Round 2

Reviewer 1 Report

Title has been changed. While I appreciate the removal of the “cold temperatures” concept, now the title has become quite meaningless and does NOT depict anything specific about the study.

The introduction has not been changed at all and still incorporates all those problematic concepts about low temperatures which are not the actual cause of poor performance of your system. The problem is the pollutant and hydraulic overloading and it’s not clearly depicted in the introduction. Therefore, the introduction must be changed DRASTICALLY.

L94-97: this concept is rather problematic. Usually, population change does not influence the pollutant concentration but only the pollutant load and flow rate. It’s very awkward to ascribe the increase in pollutant concentration to a change in population.

Figures should be put right after their explanation and not dispersed in the text. Please check one-by-one the position of each figure

Captions to each figure need to be more detailed.

Figure 12 is to me rather problematic as I am not getting how it was obtained. I don’t see how these effluent TSS and BOD concentrations from primary and secondary treatment were obtained at all.  Have you just proposed these interventions, or have you already implemented them in the real system? Where are those values taken from? Supposing they are taken from some measurements, I would expect at the very least some uncertainty bar in the graph. To me this would be a proof that the proposed interventions would actually help the WWTP overall performance getting better. However, you have NOT explained this VERY CRUCIAL point in the text at all and I am even more reluctant than before to suggest this manuscript for publication. You could use a mathematical model. To my knowledge, even the stabilization and/or the reduction of the influent flow rate and pollutant concentration to a primary clarifier does not necessarily imply a significant improvement in its TSS removal efficiency, let alone what would happen in the next processes. At this stage, based on what you have presented in the manuscript, you are reporting an 80% TSS removal efficiency with primary clarification without intervention which is quite uncommon if you don’t use any chemical. At best and occasionally, primary clarifiers may have 70% TSS removal efficiency, usually 40-50%. Not 80%. The improvement reported in Figure 12 after interventions would be to 90%. Only 10% more TSS removal. All of this is frankly weird.

It comes to my mind that this cannot be published as a regular research article but as a “perspective” or a “hypothesis”. But in either case you should be explaining EXTENSIVELY the methodology used to derive each data point of Figure 12. Only then, after checking it properly, I could maybe accept Figure 12. Alternatively, you may do a complete resubmission simply talking about the effect of equalization or pre-aeration on wastewaters influent to primary clarifiers. To me it will be more acceptable. But the concept that these interventions will improve secondary treatment is not be proven at all based on what you report in the manuscript.

Reviewer 2 Report

Dear Authors

I believe that after your revision the article became clearer, although in my opinion there are some aspects that can be improved. Regarding to the references, we verified that they aren’t still sequentially referenced in the text (Example: on page 2 they go from reference 4 to 6 and from 7 to 9). Also, in relation to some figures (examples: figure 5,6,7...), that were not previously referenced appropriately in the text. Also, regarding some figures, in my opinion, the Y-axis is not properly identified. For example, in figure 6: you put only the units "mg/L" on the Y axis and, in my opinion, you should have put "TSS Concentration (mg/L)". In the case of some figures, in my opinion, need to be titled more appropriately. For example, the title of figure 9 should be: “Effect of the equalization operation on hourly BOD concentration” and not “Equalization of hourly pollution load/BOD5/”. Another example of a title that doesn't seem right to us is that of figure 11, shouldn't it be “The effect on BOD5 and TSS” instead of “Effect of BOD5 and TSS”? Regarding figure 11, on page 11, 1st paragraph, you refer that it reflects the effect of pre-aeration on the concentration of BOD and TSS, but for me it is still not very clear which variable was tested in the 8 experiments referred to in the aforementioned figure. I read with interest your manuscript, but in my opinion, some aspects must be corrected and clarified previous to publication.

My best regards,

Conceição Mesquita

Round 3

Reviewer 1 Report

Dear authors,

I keep on not receiving any compelling explanation about the data points of Figure 12. There is no methodology supporting those values at all, especially about the TSS removal efficiency of 80% without any interventions and with the overload problems you have. This is clearly not possible and you haven't put enough efforts at justifying it whatsoever. You have not managed to change my mind and convince me those results are reasonable.

I am really sorry but I have to recommend rejection. I have insisted to get from you explanations throughout the previous reviews but you haven't provided any. 

Suggestions about how to get this article through and have it published were provided in the previous review. 

Author Response

The treatment efficiency TSS of 80% in the primary clarifiers is not produced within this study as a result. We will cite the source in the text. It is the number provided by the Wastewater treatment company as result of their routine monitoring campaigns. We were by ourself surprised by this high number, however we decided use the number as reported by the officials.

We gladly made all suggested changes to our manuscript and thank the reviewer. Suggested references are also included in our manuscript.

Round 4

Reviewer 1 Report

It's ok.    The initial "if" of line 651 is replaced with "assuming".   Also reference has been added, like suggested: 

Borzooei, S., Zanetti, M.C., Lorenzi, E., Scibilia, G. (2017). Performance Investigation of the Primary Clarifier- Case Study of Castiglione Torinese. In: Mannina, G. (eds) Frontiers in Wastewater Treatment and Modelling. FICWTM 2017. Lecture Notes in Civil Engineering , vol 4. Springer, Cham.